# The Nuclear Dense Fine Speckled (DFS) Immunofluorescence Pattern: Not All Roads Lead to DFS70/LEDGFp75

**DOI:** 10.3390/diagnostics13020222

**Published:** 2023-01-07

**Authors:** Evelyn S. Sanchez-Hernandez, Greisha L. Ortiz-Hernandez, Pedro T. Ochoa, Michael Reeves, Nicola Bizzaro, Luis E. C. Andrade, Michael Mahler, Carlos A. Casiano

**Affiliations:** 1Center for Health Disparities and Molecular Medicine, Department of Basic Sciences, Loma Linda University School of Medicine, Loma Linda, CA 92350, USA; 2Laboratorio di Patologia Clinica, Ospedale San Antonio, Azienda Sanitaria Universitaria Integrata, 33100 Udine, Italy; 3Rheumatology Division, Department of Medicine, Federal University of Sao Paulo, São Paulo 04021-001, Brazil; 4Immunology Division, Fleury Medicine and Health Laboratory, São Paulo 04023-062, Brazil; 5Werfen Autoimmunity, San Diego, CA 92131, USA; 6Rheumatology Division, Department of Medicine, Loma Linda University School of Medicine, Loma Linda, CA 92350, USA

**Keywords:** antinuclear autoantibodies, autoimmunity, dense fine speckled pattern, DFS70, HEp-2 cells, IFA, LEDGF/p75, pseudo-DFS pattern

## Abstract

The monospecific dense fine speckled (DFS) immunofluorescence assay (IFA) pattern is considered a potential marker to aid in exclusion of antinuclear antibody (ANA)-associated rheumatic diseases (AARD). This pattern is typically produced by autoantibodies against transcription co-activator DFS70/LEDGFp75, which are frequently found in healthy individuals and patients with miscellaneous inflammatory conditions. In AARD patients, these antibodies usually co-exist with disease-associated ANAs. Previous studies reported the occurrence of monospecific autoantibodies that generate a DFS-like or pseudo-DFS IFA pattern but do not react with DFS70/LEDGFp75. We characterized this pattern using confocal microscopy and immunoblotting. The target antigen associated with this pattern partially co-localized with DFS70/LEDGFp75 and its interacting partners H3K36me2, an active chromatin marker, and MLL, a transcription factor, in HEp-2 cells, suggesting a role in transcription. Immunoblotting did not reveal a common protein band immunoreactive with antibodies producing the pseudo-DFS pattern, suggesting they may recognize diverse proteins or conformational epitopes. Given the subjectivity of the HEp-2 IFA test, the awareness of pseudo-DFS autoantibodies reinforces recommendations for confirmatory testing when reporting patient antibodies producing a putative DFS pattern in a clinical setting. Future studies should focus on defining the potential diagnostic utility of the pseudo-DFS pattern and its associated antigen(s).

## 1. Introduction

Antinuclear antibody (ANA)-associated rheumatic diseases (AARD), such as systemic lupus erythematosus (SLE), Sjögren’s syndrome, systemic sclerosis (SSc), and idiopathic inflammatory myopathies (IIM), are characterized by the presence of circulating autoantibodies to a large spectrum of nuclear and cytoplasmic antigens [1,2]. ANAs have been found to target hundreds of antigens, including double-stranded DNA (dsDNA), ribosomal P proteins (Ribo-P), RNA-associated proteins (Sm, SSA/Ro, SSB/La, U1RNP, RNA Pol III), centromere proteins (CENP-A,-B,-C, and -F), nucleolar proteins (fibrillarin, Th/To), DNA topoisomerase I (Topo-I/Scl-70), and histidyl tRNA synthetase (Jo-1). Many of these autoantibodies are key tools in precision medicine due to their clinical utility as probes for differential diagnosis of AARD [1,2]. The gold standard screening method for detection of ANAs is human epithelial HEp-2-cell-based immunofluorescence assay (IFA) [3]. Individual IFA patterns associated with most known ANAs have been catalogued by the International Consensus on Antinuclear Antibody Patterns (ICAP) (www.anapatterns.org, accessed on 2 September 2022) with the purpose of promoting harmonization in the way ANA test results are communicated by diagnostic laboratories to clinicians and helping with critical clinical decisions [3,4]. ICAP has identified 29 distinct ANA patterns, designated anti-cell (AC) patterns 1 to 29, with AC-1, for instance, corresponding to “nuclear homogeneous”, characteristic of antibodies to dsDNA, nucleosomes, or histones, and AC-29 being characteristic of antibodies to Topo-I/Scl-70 [3,4,5].

The AC-2 pattern, characterized by nuclear dense fine speckles (DFS) with strong metaphase chromatin staining, is produced by autoantibodies to a chromatin-associated protein of 70 kD designated DFS70 [6,7,8]. This protein is also known as lens epithelium derived growth factor p75 (LEDGF/p75), transcription co-activator p75 (TCP75), and PC4 and SRSF1 interacting protein 1 (PSIP1) [9,10]. DFS70/LEDGFp75 has recently emerged as a multi-functional, stress-survival, transcription-associated protein with relevance not only to autoimmunity but also to cancer, acquired immunodeficiency syndrome (AIDS), and eye diseases [9,10,11,12,13,14]. Acting as a hub for protein–protein interactions in the nucleus, DFS70/LEDGFp75 facilitates tethering of transcription factors, chromatin remodelers, and other regulators to RNA polymerase II (RNAPII) transcription complexes in active chromatin sites [15,16]. This function enhances expression of stress survival and cancer-related genes, regulates mRNA splicing, promotes DNA repair via homologous recombination, and facilitates integration of human immunodeficiency one (HIV-1) virus into the chromatin of host cells [15,16,17,18,19,20,21]. DFS70/LEDGFp75 is upregulated in response to environmental stressors associated with increased oxidative stress, including chemotherapeutic drugs, nutrient deprivation, radiation, human papilloma virus (HPV) infections, and inflammatory cytokines [9,10,22]. It is unclear, however, if these stressors, particularly a pro-inflammatory tissue microenvironment, contribute to elicitation of anti-DFS70/LEDGFp75 antibodies.

Anti-DFS70/LEDGFp75 antibodies have been the subject of intense investigation in recent years due to their unusually low correlation with an AARD diagnosis or systemic autoimmunity in general. They have been detected at high titers in healthy individuals (HI) and in patients with miscellaneous inflammatory conditions, including patients with atopic disorders and undifferentiated connective tissue disease (UCTD), but at relatively low titers and frequencies in patients with AARD [23,24,25,26,27,28,29,30,31]. Typically, when present in patients with AARD, these antibodies occur together with other disease-associated ANAs, which prevents clear visualization of the DFS pattern; however, in HI and patients with non-AARD conditions, these antibodies are commonly found as monospecific or isolated (i.e., only detectable serum ANA by HEp-2 IFA) [24,25,29,31,32,33]. This has led to the hypothesis that monospecific anti-DFS70/LEDGFp75 antibodies could be used as biomarkers to aid in exclusion of AARD diagnosis in persistently ANA-positive individuals [34,35,36]. Algorithms have been proposed to use these antibodies in this capacity in clinical settings [37,38,39].

The AC-2 nuclear DFS IFA pattern, often confused with the AC-1 and AC-29 patterns, is described by ICAP as a “speckled pattern distributed throughout the interphase nucleus with characteristic heterogeneity in the size, brightness and distribution of the speckles. Throughout the interphase nucleus, there are some denser and looser areas of speckles (very characteristic feature). The metaphase plate depicts strong speckled pattern with some coarse speckles standing out.” Although this description is highly accurate and sufficiently descriptive, detection of the AC-2 pattern can still be very challenging in the clinical laboratory, particularly if the IFA technologist is not well-familiarized with the peculiarities of this pattern, automated IFA detection systems are used, or the pattern is analyzed from clinical-lab-generated microscopy images [40,41,42,43,44,45,46,47,48]. Bentow et al. [48] conducted a survey of 230 IFA technologists and reported that the AC-2 pattern was very difficult to identify when it occurred together with other ANA patterns in the same serum, and that the monospecific AC-2 pattern was often confused with the AC-1 pattern or other nuclear speckled patterns. This confusion could be influenced by the source of HEp-2 substrate used, level of expertise in ANA detection, and the resolution of the microscope. Another potential source of confusion, pointed out by Bizzaro et al. [42], is the possibility that other ANAs may be responsible for producing DFS IFA patterns resembling AC-2. This is plausible considering that DFS70/LEDGFp75 closely interacts with many chromatin-associated proteins, some with similar molecular weight and nuclear localization patterns (e.g., MeCP2), which could be targets of autoantibodies producing the DFS pattern [15,16,17,18,49,50].

Given the subtle differences between the classic or genuine DFS IFA pattern reactive with DFS70/LEDGFp75 and the “DFS-like” patterns produced by other autoantibodies, we recently proposed inclusion in ANA IFA pattern classification of a “pseudo-DFS” pattern [36,46,51,52]. Autoantibodies producing this DFS-like or pseudo-DFS pattern typically react negatively with DFS70/LEDGFp75 by chemiluminescence immunoassay (CLIA) and immunoblotting and produce by conventional fluorescence microscopy weaker metaphase staining and less heterogeneous fine speckles [36,52]. While this pattern could be considered similar to other previously described DFS-like patterns, such as the “quasi-homogenous” pattern [33], distinguishing it from the DFS70/LEDGFp75 pattern can be difficult. The pseudo-DFS pattern has not been fully characterized at high resolution microscopy and in the context of DFS70/LEDGFp75 biology.

In this study, we used confocal microscopy and immunoblotting to characterize ANAs producing a pseudo-DFS IFA pattern with no evidence of reactivity against DFS70/LEDGFp75. We also explored if this pseudo-DFS IFA pattern could be produced by autoantibodies to known interacting partners of DFS70/LEDGFp75. Our results indicate that not all ANAs producing the monospecific DFS IFA pattern react exclusively with DFS70/LEDGFp75. This raises the possibility that, while DFS70/LEDGFp75 is preferentially associated with the AC-2 pattern [49], other members of its protein interactome or the RNAPII transcription machinery could also be targeted by autoantibodies generating either this pattern or the pseudo-DFS pattern.

## 2. Materials and Methods

### 2.1. Cell Lines

Prostate cancer cell line PC3 was acquired from the American Type Culture Collection (ATCC, Manassas, VA, USA, Cat# ATCC-CRL-1435) and cultured in RPMI-1640 medium (Genesee Scientific, San Diego, CA, USA, Cat# 25-506), supplemented with 10% (*v*/*v*) fetal bovine serum (Genesee Scientific, San Diego, CA, USA, Cat# 25-514), penicillin/streptomycin (Fisher Scientific, Pittsburgh, PA, USA, Cat# 30-002-CI), and normocin 1G (Fisher Scientific, Cat# NC9390718). Cells were grown under 5% CO_2_ at 37 °C. A docetaxel-resistant PC3 cell line (PC3-DR) was developed as indicated previously [50] and maintained in the presence of 10 nM docetaxel (LC Laboratories, Woburn, MA, USA, Cat# D-1000). This PC3-DR cell line was recently shown by our group to upregulate DFS70/LEDGFp75 and members of its protein interactome and is an excellent model for evaluating the immunoreactivity of antibodies to these proteins [50]. Short tandem repeat (STR) service provided by ATCC (Cat# ATCC-135-XV) was used to authenticate the PC3 and PC3-DR cell lines. Mycoplasma testing was conducted at least twice a year using the MycoAlert^TM^ Mycoplasma Detection Kit (Lonza, Walkersville, MD, USA, Cat# LT07-218). Cells that were found contaminated were discarded. Cell lines were grown until approximately 20 passages.

### 2.2. Antibodies

De-identified human serum samples containing ANA displaying the characteristic monospecific DFS IFA pattern were from serum banks in San Diego, CA, USA (M. Mahler), Sao Paulo, Brazil (L.E.C. Andrade), and Tolmezzo, Italy (N. Bizzaro). We also used in this study commercially available rabbit antibodies to DFS70/LEDGFp75 (Bethyl Laboratories/ Fortis Life Sciences, Montgomery, TX, USA, Cat# A300-848A), Menin (Bethyl Laboratories/Fortis Life Sciences, Cat# A300-105A), JPO2 (Bethyl Laboratories/Fortis Life Sciences, Cat# A300-846A), HRP2 (Bethyl Laboratories/Fortis Life Sciences, Cat# A304-314A), IWS1 (Cell Signaling Technology, Danvers, MA, USA, Cat#5681), c-MYC (Cell Signaling Technology, Cat# 18583), PogZ (Aviva Systems Biology, San Diego, CA, USA Cat# RP39173-P050), H3K36me2 (Cell Signaling Technology, Cat# 2901T), and MLL (Bethyl Laboratories/Fortis Life Sciences, Cat# A300-086A). In addition, we used the following secondary antibodies for IFA: goat anti-human IgG (H + L) FITC (ThermoFisher Scientific, Waltham, MA, USA, Cat# 62-711), goat anti-rabbit IgG (H + L) Rhodamine (Millipore Sigma, St. Louis, MO, USA, Cat# AP 124R), and goat anti-Rabbit IgG (H + L) Alexa Fluor 488 (ThermoFisher Scientific, Cat# A-11008).

### 2.3. Screening of Human Sera for Antibodies Producing the DFS IFA Pattern

Serum samples were examined by the HEp-2 IFA test using the NOVA Lite HEp-2-ANA kit/substrate slides (Inova Diagnostics, San Diego, CA, USA) following the manufacturer’s protocol and as previously described [49]. Sera were also evaluated for the presence of anti-DFS70/LEDGFp75 antibodies using the QUANTA Flash DFS70 chemiluminescence assay (CLIA, Inova Diagnostics) as previously described [49]. The HEp-2 DFS IFA patterns displayed by selected CLIA anti-DFS70-positive and CLIA anti-DFS70-negative sera were evaluated by conventional immunofluorescence microscopy using a Keyence BZ9000 Biorevo microscope or by confocal microscopy using a Zeiss LSM-710-NLO microscope with a 63X oil immersion objective with corresponding filters. ImageJ software was used to analyze IFA images. In some experiments, selected CLIA anti-DFS70-positive and CLIA anti-DFS70-negative sera presenting the DFS IFA pattern were immunoadsorbed with a recombinant peptide corresponding to the entire DFS70/LEDGFp75 autoepitope region, which is encompassed by the C-terminal integrase binding domain (IBD, residues 349–435) of this protein [53], as previously described [49,50]. Briefly, sera were diluted at 1:80 in PBS and incubated for 2–4 h in the presence and absence of the IBD peptide. Immunostaining in HEp-2 slides was then performed to determine if the DFS IFA pattern was abolished as expected for monospecific DFS70/LEDGFp75 sera [49,50]. In other experiments, the IFA patterns generated by commercial antibodies recognizing DFS70/LEDGFp75 or selected interacting protein partners were examined in HEp-2 cells alone or co-incubated with selected CLIA anti-DFS70-positive and CLIA anti-DFS70-negative sera. Both human and rabbit antibodies were used at 1:80 dilutions. FITC or rhodamine-conjugated secondary antibodies were used at 1:100 dilutions.

### 2.4. Immunoblotting

Total protein extracts from PC3-DR cells were separated by SDS-PAGE (NuPAGE 4–12%, ThermoFisher Scientific) followed by transfer to iBlot Gel Transfer Stacks PVDF (Invitrogen Cat# IB24002) in the Invitrogen iBlot 2 transfer system. Membranes were blocked overnight with 5% dry milk in TBS-T buffer (20 mM Tris–HCl, pH 7.6, 140 mM NaCl, 0.2% Tween 20) and then probed with individual human sera or commercial antibodies for 2 h. Membranes were washed at least three times with TBS-T and then incubated for 2 h with the respective horseradish peroxidase (HRP)-conjugated secondary antibodies (goat anti-human, Invitrogen, Cat# A18847; goat anti-rabbit, Cell Signaling Technology, Cat# 7074S). After several washes with TBS-T, detection of antibodies bound to proteins was achieved by enhanced chemiluminescence (Fisher Scientific, Cat# PI34580). Protein bands were captured in X-ray films, and images of Western blots were generated by scanning the films in an Epson WF2830 scanner.

## 3. Results

### 3.1. Selected CLIA Anti-DFS70-Negative Sera Produce a Pseudo-DFS IFA Pattern Resembling That of Anti-DFS70/LEDGFp75 Autoantibodies

Our first cohort of human sera with antibodies producing the DFS IFA pattern in HEp-2 cells included 96 sera (from Andrade and Mahler, Sao Paulo and San Diego, respectively). Of these, 56 were positive for DFS70/LEDGFp75 by CLIA (cut off value of 20), with an average value of 188.99 CU. There were 40 sera with negative CLIA values (average 3.99 CU) that presented a suspected DFS-like pattern or pseudo-DFS pattern. From these two groups, we selected for further analysis five CLIA anti-DFS70-positive sera and five CLIA anti-DFS70-negative sera that presented a nuclear pattern classified as DFS (AC-2) or pseudo-DFS, with no additional pattern in the HEp-2 IFA test (Figure 1). The selected CLIA anti-DFS70-positive sera (PL11, PL22, PL48, PL61, and PL83) clearly showed, by confocal microscopy analysis, classical DFS70/LEDGFp75 antibody immunostaining characterized by DFS nuclear staining excluding the nucleoli and bright metaphase chromatin staining (Figure 1A). This was more evident for PL48, a high-titer DFS70/LEDGFp75 serum we used in our previous study to co-precipitate the IBD interacting partners of this autoantigen [50]. The selected CLIA anti-DFS70-negative sera (PL24, PL30, PL72, PL85, and PL95) presented what we considered a pseudo-DFS pattern, with serum PL72 showing increased bright speckles in the nucleus surrounding the nucleoli and PL95 showing relatively weak metaphase staining compared to interphase nuclei (Figure 1B). Sera PL24, PL30, and PL85 produced DFS patterns that were very similar, but not identical, to those of DFS70/LEDGFp75 sera (Figure 1B).

To rule out the possibility that sera producing the pseudo-DFS pattern could still be reacting with DFS70/LEDGFp75 despite negative results in the CLIA test, we pre-absorbed selected sera with an autoepitope peptide corresponding to the entire C-terminal IBD (HIV integrase binding domain) region of this autoantigen. The IBD is a hub of protein–protein interactions and serves as the binding site for the HIV-1 integrase and many transcription factors linked to cell survival signaling and oncogenesis [15,16,18,50]. Ogawa et al. [53] previously showed that anti-DFS70/LEDGFp75 antibodies react with multiple linear and conformational epitopes within the IBD, which encompasses an alpha-helical structure spanning approximately 86 amino acids (residues 349-435). As expected, immunoadsorption of selected CLIA anti-DFS70-positive sera abolished (PL11, PL48) or reduced (PL61) their DFS IFA pattern in HEp-2 cells (Figure 2A). However, this was not the case for CLIA anti-DFS70-negative sera PL24, PL72, and PL95, suggesting that their antibodies, associated with the pseudo-DFS pattern, were not targeting DFS70/LEDGFp75 (Figure 2B).

To ascertain if the classic DFS pattern produced by anti-DFS70/LEDGFp75 antibodies co-localized with the pseudo-DFS pattern, we co-incubated the CLIA anti-DFS70-negative sera PL24 and PL95 (green) with a commercial monospecific rabbit antibody to DFS70/LEDGFp75 (red) using the standard HEp-2 test and performed confocal microscopy. As a control, we evaluated the co-localization of the CLIA anti-DFS70-positive sera PL11 and PL61 with the rabbit anti-DFS70/LEDGFp75 antibody (Figure 3). As expected, the DFS pattern produced by the CLIA anti-DFS70-positive sera PL11 and PL61 showed substantial nuclear and metaphase chromosome co-localization with the anti-DFS70/LEDGFp75 rabbit antibody (yellow/orange fluorescence in the merged image, Figure 3A). In contrast, the pseudo-DFS pattern produced by CLIA anti-DFS70-negative sera PL24 and PL95 also showed areas of co-localization, but these appeared to be less extensive, as evidenced by the increased separation of the red and green fluorescence in the merged image (Figure 3B). For a more detailed visualization of this separation, we enlarged the fields corresponding to the PL61 and PL24 merged images (Figure 4). The enlarged image clearly shows substantial co-localization of the patterns (yellow color) produced by serum PL61 and the anti-DFS70/LEDGFp75 rabbit antibody (Figure 4A). However, there is less co-localization of the patterns produced by serum PL24 and the anti-DFS70/LEDGFp75 rabbit antibody given the more extensive separation of the red and green fluorescence (Figure 4B).

Interestingly, we observed that the CLIA anti-DFS70-negative PL85 serum (shown in Figure 1B) showed almost complete co-localization of its pseudo-DFS target antigen with DFS70/LEDGFp75 targeted by the rabbit serum, as evidenced by the extensive yellow fluorescence in the merged image, with very little green or red fluorescence (Figure 5A). Intriguingly, immunoadsorption of PL85 serum with the IBD peptide abolished the metaphase chromosome staining (yellow arrows), which would be expected for a CLIA anti-DFS70-positive serum but not for a CLIA anti-DFS70-negative serum (Figure 5B). However, the interphase nuclear DFS staining was not abolished. To explore the possibility that serum PL85 may have a mix of antibodies, including anti-DFS70/LEDGFp75 antibodies, despite a CLIA negative result, we performed immunoblotting analysis using total proteins from the docetaxel-resistant prostate cancer cell line PC3-DR, which we previously demonstrated to overexpress DFS70/LEDGFp75 [50]. The immunoblots revealed that, while the CLIA anti-DFS70-positive serum PL61 immunoreacted strongly with the 75 kD protein band corresponding to DFS70/LEDGFp75, serum PL85 showed no visible reactivity against any protein bands (Figure 5C).

### 3.2. The Nuclear Localization Pattern of Selected Protein Interacting Partners of DFS70/LEDGFp75 Resembles the Classic DFS IFA Pattern

Our previous studies indicated that commercial antibodies to interacting partners of DFS70/LEDGFp75 produce a DFS-like or pseudo-DFS pattern that shows extensive nuclear co-localization with this autoantigen [49,50]. To explore the possibility that human sera producing the pseudo-DFS pattern may target interacting partners of DFS70/LEDGFp75, we selected a panel of commercial antibodies to individual partners of this protein that bind to its IBD region (MLL, Menin, POGZ, JPO2, c-MYC, IWS1) or PWWP domain (H3K36me2), as well as to HRP2 (also known as hepatoma-derived growth factor 2, HDGF2), a DFS70/LEDGFp75 paralog that also contains both PWWP and IBD domains [20,50]. In our recent study, we showed that all these interacting partners are also upregulated in docetaxel-resistant PCa cells [50]. Therefore, we used total protein extract from PC3-DR cells for detection of these proteins by immunoblotting and immunofluorescence using commercial antibodies. The reactivity of these antibodies with distinct protein bands in immunoblots is shown in Figure 6. Except for Menin (68 kD), none of the selected interacting partners of DFS70/LEDGFp75 migrated around the 70 kD region. Next, we evaluated the immunoreactivity of these antibodies in HEp-2 slides to determine if they produce a pseudo-DFS pattern resembling that of DFS70/LEDGFp75 antibodies (Figure 7).

While most of the interacting partners of DFS70/LEDGFp75 showed a DFS nuclear staining pattern, metaphase chromosomes were not stained (yellow arrows) in the case of Menin, POGZ, JPO2, c-MYC, and IWS1, indicating that these proteins do not interact with DFS70/LEDGFp75 in HEp-2 mitotic chromosomes (Figure 7A). The immunostaining pattern of antibodies to HRP2 was unexpected since it showed mitotic spindle staining (yellow arrows) with no DFS nuclear staining (Figure 7A). This is in contrast with our previous observation in PC3-DR cells showing strong nuclear co-localization between HRP2 and DFS70/LEDGFp75 [50]. We noticed that MLL, an IBD interacting partner of DFS70/LEDGFp75 [18], did show a pseudo-DFS pattern in interphase and mitosis (Figure 7A). H3K36me2, a methylated histone marker of active chromatin that interacts with the N-terminal PWWP domain of DFS70/LEDGFp75 and co-localizes with this protein [20,50], also produced a pseudo-DFS pattern with mitotic chromosome staining (yellow arrows) (Figure 7A). Considering that several of the commercial rabbit sera recognized interacting partners of the DFS70/LEDGFp75 IBD region, we needed to rule out the possibility that these sera also contained antibodies that could be cross-reacting with epitopes in this region. This is, however, unlikely as pre-adsorption with the IBD peptide did not abolish their IFA immunoreactivity (Figure 7B).

Given that the IFA patterns of MLL and H3K36me2 rabbit antibodies strongly resembled that of DFS70/LEDGFp75, we co-incubated these antibodies with selected CLIA anti-DFS70-positive and -negative sera. Co-incubation of anti-MLL and anti-H3K36me2 antibodies with the CLIA anti-DFS70-positive serum PL61 produced a high degree of co-localization between the target antigens, as evidenced by the extensive yellow/orange fluorescence in the merged images (Figure 8A). Similarly, there was strong co-localization between the antigen targeted by the CLIA anti-DFS70-negative pseudo-DFS serum PL24 and the antigens recognized by the anti-MLL and H3K36me2 antibodies, although the degree of co-localization appeared to be less extensive compared to the CLIA anti-DFS70-positive serum (Figure 8B).

### 3.3. CLIA Anti-DFS70-Negative Sera Presenting the Pseudo-DFS IFA Pattern Do Not Recognize Common Protein Bands in Immunoblots

The results presented above showed that commercial antibodies to specific DFS70/LEDGFp75 interacting partners (e.g., MLL and H3K36me2) produce a pseudo-DFS IFA pattern in HEp-2 slides. This led to the question of whether these two proteins are targeted by the CLIA anti-DFS70-negative autoantibodies producing the pseudo-DFS IFA pattern. To explore this possibility, we evaluated the immunoreactivity of selected CLIA anti-DFS70-positive and CLIA anti-DFS70-negative sera against total protein extract from PC3-DR cells in immunoblots (Figure 9).

The CLIA anti-DFS70-positive sera clearly reacted with the 75 kD band corresponding to DFS70/LEDGFp75. However, the CLIA anti-DFS70-negative pseudo-DFS sera reacted with diverse bands or did not react at all. These sera did not show strong reactivity to a common and clearly defined band in the two independent immunoblots shown in Figure 9. Of note, there was no common immunoreactivity by these sera in the 16–20 kD and 160–180 kD regions, where reactivity against H3K36me2 and MLL, respectively, would be expected (Figure 9A,B). The large, dark immunoreactive regions at the very top and bottom of the blot shown in Figure 9A are considered artifacts of our multi-blot system since we often observe these regardless of the serum cohort used. Notice the absence of these dark regions in the blot shown in Figure 9B.

To confirm these results, we used a separate cohort of fifteen DFS and pseudo-DFS sera (from N. Bizzaro, Italy) with available CLIA data. The selected CLIA anti-DFS70-positive and anti-DFS70-negative sera from this cohort also produced nuclear speckled IFA patterns (Figure 10A). However, consistent with the results shown in Figure 9, only the CLIA anti-DFS70-positive sera reacted with the 75 kD band corresponding to DFS70/LEDGFp75, while the CLIA anti-DFS70-negative sera did not show strong reactivity to any common or well-defined protein band (Figure 10B). The CLIA anti-DFS70-negative sera B7 and B8 reacted with weakly diffuse bands around 100 kD that could potentially correspond to Topo-I/Scl-70 (AC-29 pattern). Supporting this possibility, the well-defined fluorescent ring around the nucleoli typically associated with anti-Topo-I/Scl-70 antibody staining [5] was visible for sera B7 and B8 using confocal microscopy. It is, therefore, possible that serum B9 is a true pseudo-DFS serum, while sera B7 and B8 could be anti-Topo-I/Scl-70 sera. These results underscore the subtleties of distinguishing the genuine DFS pattern (AC-2) from the pseudo-DFS pattern and the Topo-I/Scl-70 pattern (AC-29).

## 4. Discussion

Accurate identification of the AC-2 IFA pattern produced by autoantibodies to DFS70/LEDGFp75 is crucial for their clinical use as potential biomarkers to aid in the exclusion of AARD. While the presence of DFS70/LEDGFp75 autoantibodies per se does not rule out an AARD diagnosis, sufficient evidence has accumulated over the past decade indicating that monospecific anti-DFS70/LEDGFp75 reactivity is seldom observed in AARD [22,23,24,25,26,27,28,29,30,31,32,33,34]. As indicated by Bentow et al. [48], the documented confusion of the AC-2 pattern with other nuclear IFA patterns, such as those produced by antibodies against dsDNA (AC-1, homogeneous), SSA/Ro (AC-4, nuclear fine speckled), Sm and other ribonucleoproteins (AC-5, nuclear large/coarse speckles), and Scl-70/Topo-I (AC-29), could lead to costly and excessive diagnostic testing, potentially delaying a positive or negative diagnosis of AARD. This may also produce unnecessary anxiety in patients and their families who are not familiar with the diagnostic implications of a positive DFS ANA pattern. Because of this caveat and other challenges associated with accurate detection of monospecific DFS70/LEDGFp75 autoantibody reactivity as judged by the HEp-2 IFA test, members of our team and others have emphasized that definite identification of these antibodies may not rely solely on this test and should involve confirmatory tests, such as CLIA, ELISA, immunoblotting, and line immunoassays as additional and necessary detection tools in the clinical setting [41,42,43,44,45,49,54,55,56,57]. This was proven to be a critical step when applying use of the monospecific DFS70/LEDGFp75 autoantibody to a precision medicine approach involving individualized immunodiagnostics [58].

In previous studies, Bizzaro et al. [41,42] reported that not all serum samples displaying the presumptive DFS IFA pattern in HEp-2 cells could be confirmed to be anti-DFS70/LEDGFp75 with CLIA or other immunoassays. These differences could be attributed to misinterpretation of other IFA patterns, particularly AC-1 and AC-29, use of different immunoassay platforms, or the very nature of the DFS70/LEDGFp75 antigen (truncated versus full-length, denatured versus native) used in the different immunoassays. These investigators did not rule out the possibility that certain sera producing the DFS IFA pattern may contain antibodies to proteins that form complexes with DFS70/LEDGFp75 in the nucleus [42]. We previously explored this possibility by examining whether MeCP2, a 70 kD chromatin-associated protein that interacts with the N-terminal PWWP domain of DFS70/LEDGFp75, could also be a target of autoantibodies producing the DFS pattern [48]. However, despite identical migration of MeCP2 and DFS70/LEDGFp75 in immunoblots and their extensive nuclear co-localization, human sera producing the monospecific DFS pattern reacted preferentially with DFS70/LEDGFp75, with excellent correlation between the HEp-2 IFA test, CLIA, immunoblotting, and ELISA [23,25,28,49].

Despite these results, the possibility that the monospecific DFS pattern may not be exclusively associated with antibodies to DFS70/LEDGFp75 is conceivable given the emerging evidence from our group and others that this protein is engaged in protein–protein interactions, both through its N-terminal PWWP and C-terminal IBD domains, with multiple proteins associated with regulation and execution of a wide variety of cellular functions, including RNAPII transcription, cellular antioxidant and stress survival, malignant transformation, mRNA splicing, DNA repair, and chromatin remodeling [15,16,17,18,19,20,21,50]. In a recent study, we showed that some of these interacting partners, particularly those associated with the IBD domain, are co-upregulated and form complexes in the nucleus of PCa cells that have transitioned to docetaxel resistance [50]. These DFS70/LEDGFp75 protein complexes appear to be functionally critical since their disruption by gene depletion leads to decreased survival, clonogenicity, and tumorsphere formation of the docetaxel-resistant cells [50]. Therefore, it could not be ruled out that immune recognition of DFS70/LEDGFp75 leading to generation of autoantibodies may also spread to other proteins that are structurally and functionally associated with this autoantigen. Yet, evidence for this notion remains elusive.

In the present study, we used confocal microscopy to compare at higher resolution the genuine DFS pattern produced by anti-DFS70/LEDGFp75 sera in HEp-2 cells with the pseudo-DFS pattern produced by sera that present a CLIA anti-DFS70-negative result. Our study was designed to investigate three aims: (1) determine the degree of resemblance of the DFS IFA pattern produced by anti-DFS70/LEDGFp75 sera with the pseudo-DFS pattern produced by CLIA anti-DFS70-negative sera; (2) determine if known interacting partners of DFS70/LEDGFp75 also produce the DFS pattern in the HEp-2 IFA test; and (3) assess if these partners or other antigens are common targets of CLIA anti-DFS70-negative sera presenting the pseudo-DFS pattern.

Our results confirmed the presence of serum autoantibodies that, despite testing negative by CLIA, produce a pseudo-DFS IFA pattern that resembles that of human anti-DFS70/LEDGFp75 autoantibodies. These pseudo-DFS autoantibodies do not appear to target the IBD region of DFS70/LEDGFp75 since immunoadsorption experiments with the IBD autoepitope peptide failed to abolish their immunoreactivity. An exception was serum PL85, which, despite testing negative by CLIA and immunoblotting, showed extensive co-localization with DFS70/LEDGFp75 and lost mitotic chromosome staining, but not nuclear DFS staining, in the presence of the IBD peptide. It is possible that this serum contains antibodies to a conformational epitope within the DFS70/LEDGFp75 IBD that is detectable in mitotic HEp-2 cells but not detectable by CLIA or immunoblotting. This would be consistent with the report by Ogawa et al. [53] showing that anti-DFS70/LEDGFp75 antibodies react with a predominantly conformational epitope that encompasses the entire 86-residue IBD region (residues 349–435).

Our confocal microscopy analysis revealed that while DFS70/LEDGFp75 targeted by human autoantibodies presents a high degree of co-localization with DFS70/LEDGFp75 targeted by commercially available rabbit antibodies, the pseudo-DFS autoantibody pattern only showed partial co-localization with this protein. In our recent study, we showed that while DFS70/LEDGFp75 co-immunoprecipitated with the interacting partners examined in Figure 9, their co-localization was partial, most likely restricted to areas of transcriptionally active chromatin containing RNAPII complexes [50]. This raised the possibility that the antibodies producing the pseudo-DFS pattern may recognize interacting partners of DFS70/LEDGFp75 present in these complexes. Our analysis of the immunoreactivity of rabbit antibodies to selected DFS70/LEDGFp75 interacting partners revealed that, while most of these antibodies do not stain metaphase chromosomes, the antibodies to MLL and H3K36me2 produced a DFS-like pattern that partially co-localized with both the genuine anti-DFS70/LEDGFp75 and the pseudo-DFS pattern (Figure 8). However, immunoblotting analysis of sera producing the pseudo-DFS pattern did not reveal a clearly defined immunoreactive protein band (Figure 9), consistent with a similar observation made by Mariz et al. [33] during characterization of autoantibodies producing the nuclear fine speckled pattern. We cannot rule out that sera producing the pseudo-DFS pattern react with MLL given the high molecular weight of this protein (>300 kD), which is beyond the detection range of our immunoblotting system. The commercial rabbit antibody to MLL used in our study recognized a protein band near the 150–200 kD region (Figure 6), which may correspond to a degradation product of this protein. This antibody was used in our previous study to demonstrate the co-localization and co-immunoprecipitation of MLL with DFS70/LEDGFp75 in docetaxel-resistant PCa cells [50]. However, while one CLIA-negative serum, PL13, reacted with a protein band around 150 kD (Figure 9A), this band was not common to the other pseudo-DFS sera. Future studies could explore if knocking down MLL abolishes this reactivity.

The absence of a clearly defined common protein band recognized by sera producing the pseudo-DFS pattern also suggested the possibility that these sera may not recognize a single specific protein but several heterogenous proteins that could be associated with transcriptionally active chromatin and RNAPII complexes. We should note, however, that, except for the active chromatin marker H3K36me2, we limited our analysis of interacting partners of DFS70/LEDGFp75 to selected proteins that bind to the autoepitope IBD region. The N-terminal PWWP domain of DFS70/LEDGFp75 also interacts with several proteins involved in regulation of mRNA splicing and DNA repair [9,10,17,19,49]. These proteins co-localize in the nucleus with DFS70/LEDGFp75 and could potentially be targets of antibodies producing the pseudo-DFS pattern.

An intriguing observation in our analysis of the staining pattern of interacting partners of DFS70/LEDGFp75 in HEp-2 cells was the mitotic spindle pattern produced by antibodies to HRP2. In our previous study, we showed, using the same antibodies used in the present study, that DFS70/LEDGFp75 and HRP2 co-immunoprecipitated and co-localized in the nucleus of docetaxel-resistant PCa cells [50]. While there is no evidence that these two proteins physically interact, they are considered paralog proteins with similar, albeit most likely redundant, roles in transcription and HIV integration [59,60,61]. DFS70/LEDGFp75 and HRP2 are the only known cellular proteins that have both N-terminal PWWP and C-terminal IBD domains [60]. Thus, we expected HRP2 (~100–120 kD) to be the most likely candidate to be recognized by sera producing the pseudo-DFS pattern, and even by some anti-DFS70/LEDGFp75 sera, given that its IBD region could be cross-targeted by anti-DFS70/LEDGFp75 autoantibodies. Only two CLIA anti-DFS70-negative sera, PL24 and PL28, reacted with a band around 100 kD in one of the blots we performed (Figure 9A). However, this band was not common to other CLIA anti-DFS70-negative sera. It would be of interest in future studies to determine if knockdown of HRP2 abolishes this band. Intriguingly, our immunoblotting analysis also revealed that several CLIA anti-DFS70-positive sera (PL61 and PL63) also reacted with a clearly defined band in the ~100–120 kD vicinity (Figure 9A). Future knockdown studies could also reveal if this band corresponds to HRP2. In addition, the observed lack of mitotic chromatin staining by the anti-HRP2 antibodies in the HEp-2 cells is consistent with exclusion of this protein from mitotic chromatin and its localization to areas around the separating chromosomes in other cell types [62]. Further studies are warranted to determine if human anti-DFS70/LEDGFp75 autoantibodies cross-react with the IBD region of HRP2.

A limitation of this study is that the clinical significance of the pseudo-DFS autoantibodies, similar to that of the DFS70/LEDGFp75 autoantibodies, is still unclear. A preliminary assessment of clinical disorders that were present in patients producing the pseudo-DFS antibodies revealed miscellaneous inflammatory conditions, including celiac disease, rheumatoid arthritis, osteoarthrosis, and endometriosis. Future studies should include an exhaustive analysis of clinical conditions associated with the pseudo-DFS pattern in large patient cohorts.

## 5. Conclusions

The present study was designed to characterize in more detail the pseudo-DFS or DFS-like IFA pattern, which has been reported as a common occurrence in studies related to anti-DFS70/LEDGFp75 antibodies [33,36,40,41,42,44,46,51,52]. Our results showed that the nuclear autoantigen associated with the pseudo-DFS pattern co-localizes partially with DFS70/LEDGFp75 and its interacting partners MLL and H3K36me2, suggesting that this antigen may play a role in the biological process that occurs within active chromatin (e.g., transcription, mRNA splicing, chromatin remodeling, DNA damage repair). Other selected interacting partners of DFS70/LEDGFp75 were excluded from the mitotic chromatin in HEp-2 cells and most likely are not targets of autoantibodies producing the pseudo-DFS pattern. We were not able to identify by immunoblotting a clearly defined common immunoreactive protein band targeted by these autoantibodies, suggesting that they may react with a diverse group of proteins of various molecular weights, including very large or small proteins that are beyond the detection limits of our immunoblotting system, or recognize conformational epitopes not detectable by immunoblotting. Occurrence of serum autoantibodies that produce the pseudo-DFS pattern but are not reactive against DFS70/LEDGFp75 must be carefully considered in studies focused on prevalence of antibodies producing the DFS pattern in different populations. It should also serve as a strong justification for mandatory confirmatory tests (e.g., CLIA, ELISA, line immunoassays, or immunoblotting) when the presence of anti-DFS70/LEDGFp75 autoantibodies is reported in a clinical laboratory setting. This would be particularly relevant when patient antibodies producing the DFS pattern are used in a precision medicine approach involving individualized immunodiagnostics testing to assist in exclusion of AARD. Finally, future multicenter studies with large patient cohorts should focus on more exhaustive identification of target antigen(s) recognized by antibodies producing the pseudo-DFS pattern and their clinical significance. This will be crucial for considering inclusion of the pseudo-IFA pattern within the ICAP ANA pattern classification system.

## Figures and Tables

**Figure 1 diagnostics-13-00222-f001:**
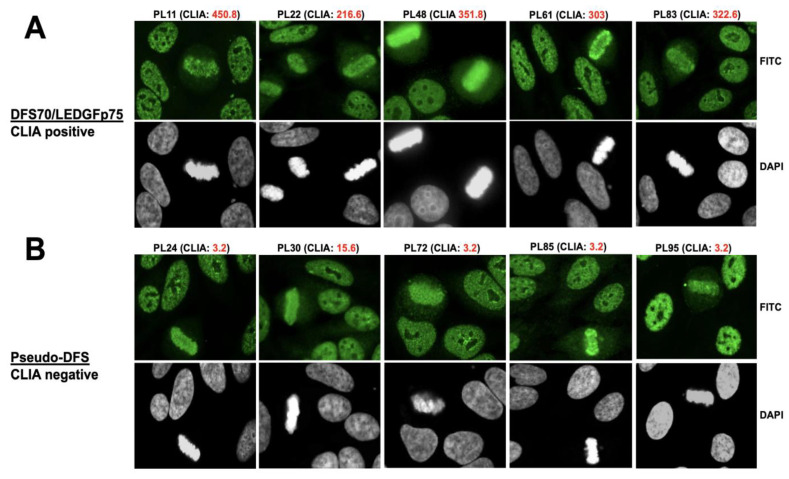
Indirect immunofluorescence images of human sera presenting the monospecific nuclear dense fine speckled (DFS) pattern. (**A**) Selected sera positive by QUANTA Flash DFS70 chemiluminescence assay (CLIA) show the DFS IFA pattern corresponding to DFS70/LEDGFp75 antibodies in HEp-2 cells. (**B**) Selected CLIA anti-DFS70-negative sera showing a pseudo-DFS IFA pattern in HEp-2 cells. Images were obtained by confocal microscopy. Chromatin staining with DAPI is presented in black and white for better visualization.

**Figure 2 diagnostics-13-00222-f002:**
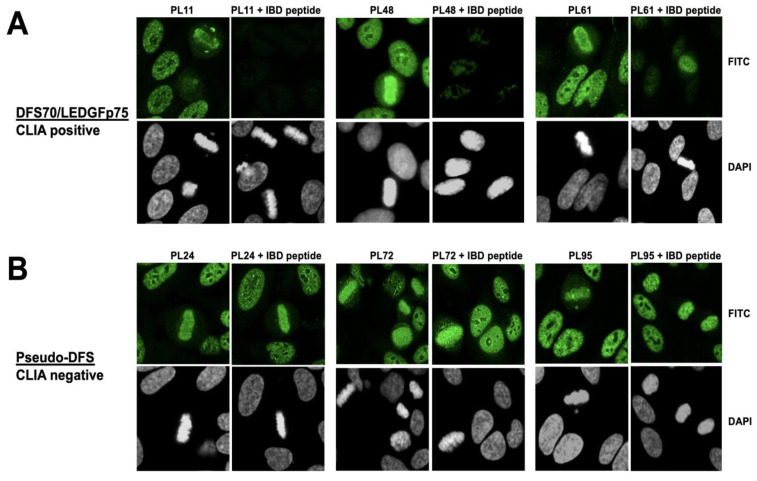
Immunoadsorption of sera with the DFS70/LEDGFp75 integrase binding domain (IBD) autoepitope peptide. (**A**) Pre-absorption of CLIA anti-DFS70-positive sera with the IBD peptide abolished their DFS IFA staining in HEp-2 cells. (**B**) Pre-absorption of CLIA anti-DFS70-negative sera with the IBD peptide failed to abolish their pseudo-DFS IFA staining. Images were obtained by confocal microscopy. Chromatin staining with DAPI is presented in black and white for better visualization.

**Figure 3 diagnostics-13-00222-f003:**
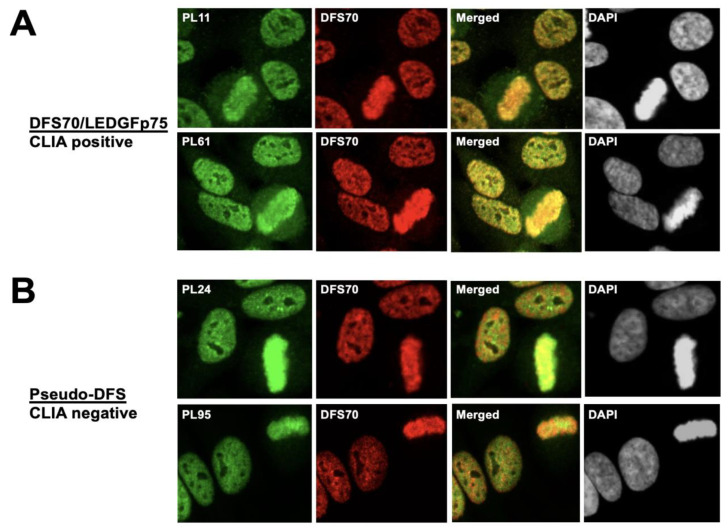
Co-localization of IFA patterns of monospecific CLIA anti-DFS70-positive and CLIA anti-DFS70-negative human antibodies with the DFS pattern produced by anti-DFS70/LEDGFp75 rabbit antibodies. (**A**) CLIA anti-DFS70-positive sera PL11 and PL61 (green) producing the classical DFS IFA pattern were co-incubated with a rabbit antibody to DFS70/LEDGFp75 (red) in the standard HEp-2 test. Extensive co-localization between the DFS70/LEDGFp75 proteins recognized by the human autoantibodies and the rabbit antibody can be observed, as evidenced by the yellow/orange fluorescence in the merged images. (**B**) Monospecific CLIA anti-DFS70-negative sera PL24 and PL95 (green) were co-incubated with a rabbit antibody to DFS70/LEDGFp75 (red). While areas of co-localization between the human pseudo-DFS antibodies and the rabbit anti-DFS70/LEDGFp75 antibody can be detected (yellow/orange fluorescence), this co-localization is not as extensive as that obtained with CLIA anti-DFS70-positive sera (shown in panel (**A**)) given the increased separation of red and green fluorescence in the merged images. Images were obtained by confocal microscopy. Chromatin staining with DAPI is presented in black and white for better visualization.

**Figure 4 diagnostics-13-00222-f004:**
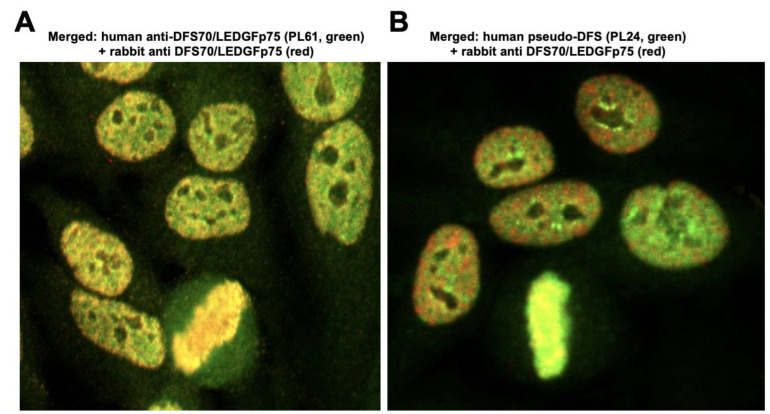
Enlarged fields showing the co-localization of the IFA patterns produced by the monospecific CLIA anti-DFS70-positive serum PL61 and CLIA anti-DFS70-negative serum PL24 with the DFS pattern produced by anti-DFS70/LEDGFp75 rabbit antibodies. (**A**) Merged image of serum PL61 (green) co-incubated with anti-DFS70/LEDGFp75 rabbit antibody (red) in the standard HEp-2 test. Extensive co-localization can be observed, as evidenced by the yellow/orange fluorescence. (**B**) Merged image of serum PL24 (green) co-incubated with anti-DFS70/LEDGFp75 rabbit antibody (red). The co-localization was less extensive, as evidenced by the increased separation of the red and green fluorescence. Images were obtained by confocal microscopy.

**Figure 5 diagnostics-13-00222-f005:**
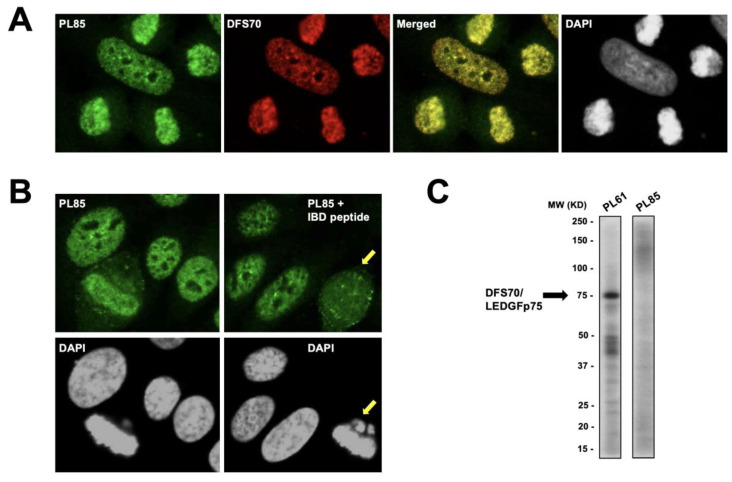
Analysis of CLIA anti-DFS70-negative serum PL85. (**A**) The monospecific pseudo-DFS IFA pattern from serum PL85 (green) shows extensive co-localization with the IFA pattern produced by the DFS70/LEDGFp75 rabbit antibody (red) in HEp-2 cells as revealed by the widespread yellow fluorescence in the merged images. (**B**) Pre-adsorption of serum PL85 with IBD peptide abolished the metaphase chromosome staining (yellow arrows) but not the interphase nuclear DFS staining. Fluorescent images were obtained by confocal microscopy. Chromatin staining with DAPI is presented in black and white for better visualization. (**C**) Immunoblotting analysis (total PC3-DR proteins) of PL85 serum antibodiesfailed to reveal any defined reactivity against protein bands as compared to the distinct reactivity of CLIA anti-DFS70-positive serum PL61 with DFS70/LEDGFp75.

**Figure 6 diagnostics-13-00222-f006:**
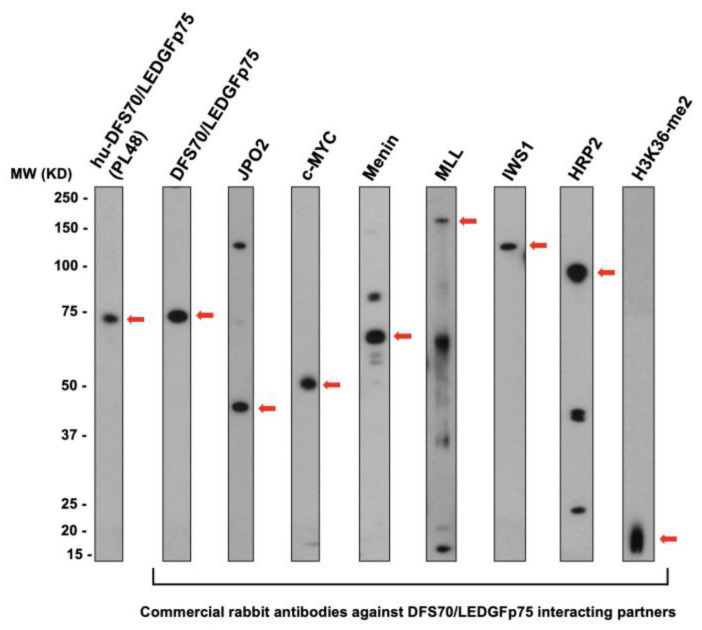
Immunoblots of selected DFS70/LEDGFp75 interacting partners. Total proteins from PC3-DR prostate cancer cells were used as substrates. Individual protein strips were incubated with commercially available rabbit antibodies against selected interacting partners of DFS70/LEDGFp75. Human anti-DFS70/LEDGFp75 serum PL48 was used as a positive control. Red arrows point to bands corresponding to the target proteins. Antibodies to Menin, MLL, and HRP2 reacted with additional bands considered non-specific or degradation products.

**Figure 7 diagnostics-13-00222-f007:**
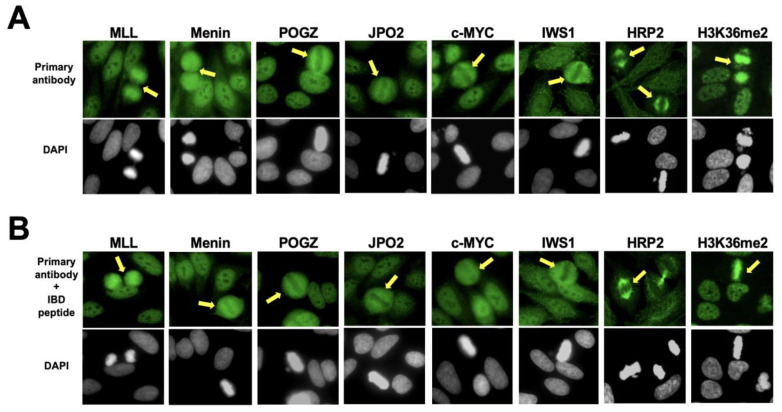
Immunofluorescence antibody patterns of selected DFS70/LEDGFp75 interacting partners in HEp-2 cells. (**A**) The IFA patterns of rabbit antibodies to the DFS70/LEDGFp75 interacting partners Menin, POGZ, JOP2, c-MYC, and IWS1 show nuclear dense fine speckles without strong and well-defined mitotic chromatin staining. HRP2 showed mitotic spindle staining without nuclear dense fine speckles. MLL and H3K36me2 showed both nuclear dense fine speckles and intense mitotic chromatin staining.Yellow arrows point to mitotic cells. (**B**) Pre-adsorption of the rabbit antibodies to DFS70/LEDGFp75 interacting partners with the IBD autoepitope peptide did not abolish their immunoreactivity, ruling out the possibility that these antibodies may be cross-reacting with the DFS70/LEDGFp75 IBD region. Fluorescent images were obtained using a Keyence Biorevo microscope. Chromatin staining with DAPI is presented in black and white for better visualization.

**Figure 8 diagnostics-13-00222-f008:**
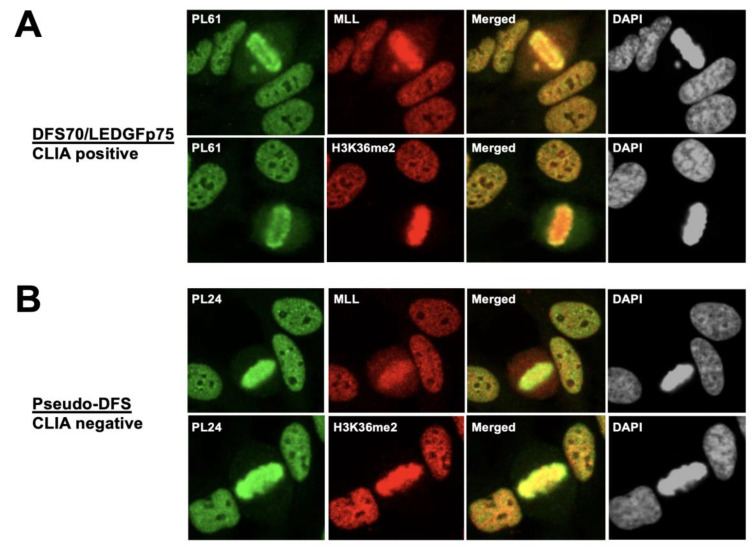
Co-localization of monospecific CLIA anti-DFS70-positive and anti-DFS70-negative autoantibodies with anti-MLL and anti-H3K36me2 antibodies. (**A**) CLIA anti-DFS70-positive anti-DFS70/LEDGFp75 serum PL61 (green) was co-incubated with rabbit antibodies to MLL or H3K36me2 (red) in the HEp-2 IFA test. Co-localization between DFS70/LEDGFp75 and its known interacting partners MLL (IBD region) and H3K36me2 (PWWP domain) is observed, as evidenced by the extensive yellow/orange fluorescence of the merged images. (**B**) CLIA anti-DFS70-negative serum PL24 was also co-incubated with rabbit antibodies to MLL or H3K36me2. Areas of strong co-localization between the pseudo-DFS pattern produced by serum PL24 and the patterns produced by the anti-MLL and H3K36me2 rabbit antibodies can be detected (yellow/orange fluorescence in merged images). However, the co-localization is not as extensive as that shown in panel A given the increased separation of red and green fluorescence in the merged images. Images were obtained by confocal microscopy. Chromatin staining with DAPI is presented in black and white for better visualization.

**Figure 9 diagnostics-13-00222-f009:**
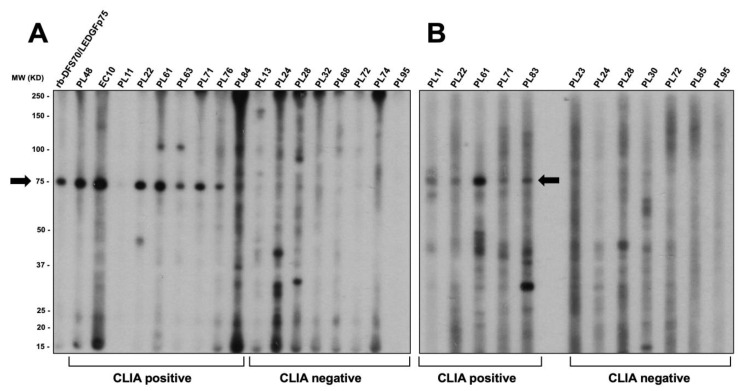
Immunoblotting analysis of selected CLIA anti-DFS70-positive and CLIA anti-DFS70-negative sera. Total proteins from PC3-DR prostate cancer cells were used as substrates in immunoblotting experiments. (**A**,**B**) show representative blots from two independent experiments. While CLIA anti-DFS70-positive sera reacted with the 75 kD band corresponding to DFS70/LEDGFp75, a common pattern of immunoreactivity could not be detected for the CLIA anti-DFS70-negative sera. Black arrows point to the DFS70/LEDGFp75 band.

**Figure 10 diagnostics-13-00222-f010:**
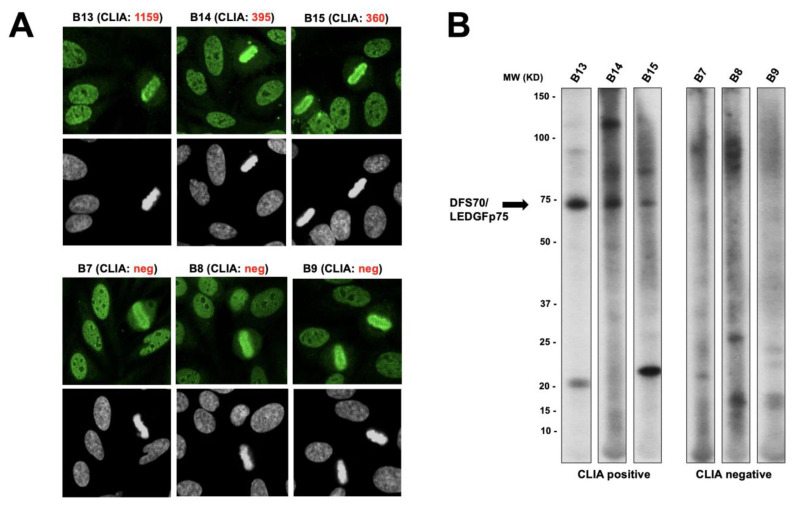
Analysis of a different cohort of CLIA anti-DFS70-positive and anti-DFS70-negative producing a speckled nuclear pattern. (**A**) The top panels show the classic DFS IFA patterns corresponding to anti-DFS70/LEDGFp75 antibodies produced by selected CLIA anti-DFS70-positive sera. The bottom panels show the pseudo-DFS IFA patterns produced by selected CLIA anti-DFS70-negative sera. Fluorescence images were obtained by confocal microscopy. Chromatin staining with DAPI is presented in black and white for better visualization. (**B**) Immunoblots of total proteins from PC3-DR prostate cancer cells were probed with selected CLIA anti-DFS70-positive and -negative sera. CLIA anti-DFS70-positive sera, but not CLIA anti-DFS70-negative sera, reacted with the 75 kD band corresponding to DFS70/LEDGFp75.

## Data Availability

The data presented here are available on request from the corresponding author.

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
