# Peer review of "The Nuclear Dense Fine Speckled (DFS) Immunofluorescence Pattern: Not All Roads Lead to DFS70/LEDGFp75"

_diagnostics, 2023, doi:10.3390/diagnostics13020222_

Round 1

Reviewer 1 Report

The authors extensively investigated and evaluated the pseudo-DFS IFA patterns of anti-nuclear antibodies. The data are clearly presented and properly interpreted and well-discussed. The clinical significance of the anti-DFS70/LEDGFp75 antibodies and pseudo-DFS IFA patterns are still unclear so far, and hence the impact of this study might not be so high. However, the study pauses an intriguing scientific question with a couple of solid data, which is worth further investigation.

Author Response

Response to Reviewer 1

Comment. The authors extensively investigated and evaluated the pseudo-DFS IFA patterns of anti-nuclear antibodies. The data are clearly presented and properly interpreted and well-discussed. The clinical significance of the anti-DFS70/LEDGFp75 antibodies and pseudo-DFS IFA patterns are still unclear so far, and hence the impact of this study might not be so high. However, the study pauses an intriguing scientific question with a couple of solid data, which is worth further investigation.

We appreciate that the reviewer found the data clearly presented and properly interpreted and well-discussed.  We also appreciate the assessment that the study poses an intriguing scientific question that is worth further investigation.  We agree with this assessment since our goal in this manuscript is to provide for the first time a careful characterization of the pseudo-DFS IFA pattern, which has been alluded in previous papers by several groups but not characterized.  We consider this a first step to stimulate further research on the autoantibodies producing this pattern.  In addition, we want to reinforce in the manuscript the notion that the DFS70/LEDGFp75 pattern could be confused with other similar patterns, therefore requiring a careful characterization or confirmatory tests before it is reported in a clinical setting.

In addition, we agree with the reviewer that because the clinical significance of the anti-DFS70/LEDGFp75 antibodies and pseudo-DFS IFA patterns are still unclear, the impact of the study might not be so high.  We have added a sentence to the Discussion (lines 815-821) recognizing this limitation.

Reviewer 2 Report

1.  Lines 307-313 are the same as lines 314-320.   One of these sections needs to be removed

2.  In Fig 3 the separation between red & green is present but it requires very careful inspection.   Do you have a picture where this separation is easier to detect?

3.  In both Fig  4B and Fig 5 explain what the yellow arrows are pointing to in the explanation of these figures.'

4.  In line 461 an "in" needs to be inserted.    "....with epitopes in this region." 

5.  In Fig 7 PL13 has a band in the 160-180 region.   Is this anti MLL?

6,  Lines 698-704.   In Fig 7A PL24 has a band in the 100KD region.   Could this represent antibody to HRP2? 

7.  I was unable to pull up and visualize S1. 

Author Response

Response to Reviewer 2

  1. Lines 307-313 are the same as lines 314-320.   One of these sections needs to be removed.

We thank the reviewer for pointing to this duplication which was an oversight on our part.  The duplicated section has been eliminated. 

  1. In Fig 3 the separation between red & green is present but it requires very careful inspection.   Do you have a picture where this separation is easier to detect?

We understand this concern and have now provided a new figure (Fig. 4) that shows an enlarged field of the merged images included in Fig. 3 for serum samples PL61 and PL24.  The new enlarged merged images clearly show substantial co-localization of the CLIA-positive serum PL61 pattern (green) with the DFS70/LEDGFp75 pattern (red), as evidenced by the abundance of yellow color in the image. By contrast, the pattern produced by the CLIA-negative pseudo-DFS serum PL24 (green) lacks extensive co-localization with the DFS70/LEDGFp75 pattern (red), as evidenced by the increased separation of the green and red fluorescence.  These results are now discussed on page 7, lines 332-338.  The new Figure 4 appears in page 8 with its corresponding legend (lines 402-410).

  1. In both Fig  4B and Fig 5 explain what the yellow arrows are pointing to in the explanation of these figures.

We thank the reviewer for pointing to this omission which has now been included in the legends of these figures (now Figures 5 and 7, respectively).  The explanation of these “yellow arrows” also appear in the Results section (lines 417, 527, 530, and 537).

  1. In line 461 an "in" needs to be inserted.    "....with epitopes in this region." 

            Thanks again for pointing to this omission, which has been inserted (line 540).

  1. In Fig 7 PL13 has a band in the 160-180 region.  Is this anti MLL?

This is a possibility that needs to be more carefully examined in future studies.  Fig. 7 is now Fig. 9. We have included the following sentence in the Discussion to address this point:    

“However, while one CLIA-negative serum, PL13, reacted with a protein band around 150 kD (Fig. 9A), this band was not common to the other pseudo-DFS sera. Future studies could explore if knocking down MLL abolishes this reactivity.” (pages 15, lines  775-778)

  1. Lines 698-704.   In Fig 7A PL24 has a band in the 100KD region.   Could this represent antibody to HRP2?

This is also another intriguing possibility that needs to be more carefully examined in future studies.  Fig. 7A is now Fig. 9A. We have included the following sentences in the Discussion to address this point:    

“Only two CLIA anti-DFS70-negative sera, PL24 and PL28, reacted with a band around 100 kD in one of the blots we performed (Fig. 9A). However, this band was not common to other CLIA anti-DFS70-negative sera.  It would be of interest in future studies to determine if knockdown of HRP2 abolishes this band. Intriguingly, our immunoblotting analysis also revealed that several CLIA anti-DFS70-positive sera (PL61 and PL63) also reacted with a clearly defined band in the ~100-120 kD vicinity (Fig. 9A). Future knockdown studies could also reveal if this band corresponds to HRP2.” (page 16, lines 803-810)

  1. I was unable to pull up and visualize S1.

We regret that the reviewer was unable to find and visualize Fig. S1.  Given the importance of this figure, and to avoid this problem to continue, we have moved Fig. S1 to the manuscript text (now Fig. 6 with corresponding legend). 
